# Effect of the Case for Carpule as a Visual Passive Distraction Tool on Dental Fear and Anxiety: A Pilot Study

**DOI:** 10.3390/ijerph20031793

**Published:** 2023-01-18

**Authors:** Nayara Bertoldo Barbosa, Bárbara Rocha Rodrigues, Isabela Ribeiro Madalena, Fernando Carlos Hueb de Menezes, César Penazzo Lepri, Maria Beatriz Carvalho Ribeiro de Oliveira, Michelle Gomides Dumont Campos, Maria Angélica Hueb de Menezes Oliveira

**Affiliations:** 1Department of Biomaterials, University of Uberaba-UNIUBE, Uberaba 38010-200, MG, Brazil; 2Department of Dentistry, University of Joinville Region-UNIVILLE, Joinville 89219-710, SC, Brazil; 3School of Dentistry, Presidente Tancredo de Almeida Neves University Center-UNIPTAN, São João del Rei 36307-251, MG, Brazil

**Keywords:** dental anxiety, local anesthesia, visual distraction, child management

## Abstract

The present cross-over study aimed to evaluate the effect of a visual passive distraction tool, a case for carpule, in the management of fear and anxiety during invasive dental treatment. Children between the ages of 4 and 8 years that need at least two treatments with inferior alveolar nerve block were selected. All the included children received two treatments at different moments: in one treatment, local anesthesia was performed with a visual passive distraction tool, the crocodile case, for carpule (experimental); in the other treatment, local anesthesia was performed without the device (control). An evaluation of the anxiety levels was performed by measuring the heart rate and salivary cortisol levels. Wilcoxon’s nonparametric test was used for a data analysis (*p* < 0.05). The mean heart rate with the visual passive distraction device fell from 81.8 to 78.1, while the control rose from 91.4 to 100 (*p* < 0.05). The mean salivary cortisol levels with the visual passive distraction device fell from 2.0 to 1.6, while in the control, they rose from 2.1 to 2.4 (*p* < 0.05). In conclusion, the crocodile case for carpule as a visual passive distraction device contributed positively to the management of fear and anxiety during inferior alveolar nerve block in children.

## 1. Introduction

Dental fear and anxiety refer to the anxiety related to dental treatment and thought of visiting a dentist for preventive or therapeutic procedures [1]. It is important to highlight that the management of dental fear and anxiety is essential for the quality and optimization of pediatric dental practice [2,3,4]. Fear and anxiety negatively contribute toward behavioral management, resulting in a stressful and unpleasant experience for both the patient and the dentist [5,6]. They can also interfere with pain thresholds during and after dental visits [7,8]. Children with exacerbated dental fear/anxiety tend to have exaggerated expectations of pain [2,7,9,10].

Behavioral management techniques for children and their families to control fear and anxiety are widely described in the literature, varying according to the child’s age and parental acceptance. The American Academy of Pediatric Dentistry (AAPD) divided behavior management techniques into two categories: Techniques with non-pharmacological and pharmacological techniques [11]. Techniques with a non-pharmacological approach such as tell-show-do, modeling, distraction, positive reinforcement, protective stabilization, and voice control are used effectively for the majority of children. Pharmacological techniques that use conscious and unconscious drug sedation to optimize management are also common in dental practice but are applied in more specific cases [2,6,12,13,14].

Non-pharmacological behavioral management techniques are simple and safe, and are preferred by parents/guardians and professionals [15,16]. Passive distraction, in particular, aims to divert the patient’s attention from what may be perceived as an unpleasant procedure. For example, techniques of visual and auditory passive distraction performed through images, cartoons, stories recorded in audio, videos, and devices that camouflage instrumentals, among others [3,4,17,18]. Behavioral management by visual and/or auditory passive distraction techniques has provided positive results for pediatric patients during various dental procedures, including procedures performed under local anesthesia [3,4,6,17,18].

The local anesthesia procedure is extensively reported as one of the most prevalent specific causes of fear/anxiety in pediatric dental practice [19,20,21]. Thus, to reduce fear and anxiety in the local anesthesia procedure using non-pharmacological techniques, a device to cover the carpule syringe and needle was developed. The “crocodile” case for carpule is made of flexible rubber, with a playful shape (crocodile) that fits and hides the carpule syringe and needle at the time of anesthesia. The null hypothesis in this study is that the crocodile case for carpule does not significantly affect fear and anxiety during the local anesthesia procedure. Therefore, the present cross-over study aimed to evaluate the influence of the crocodile case for carpule (Angie by Angelus^®^, Londrina, PR, Brazil) as a passive visual distraction device, in the management of fear and anxiety during inferior alveolar nerve block in children.

## 2. Materials and Methods

### 2.1. Ethical Aspects

This study was approved by the Ethics Committee of the School of Dentistry of Uberaba University (#0052.0.227.000-11). All parents and caregivers were appropriately clarified about this research and signed their informed consent. The study was performed in accordance with the latest version of the Declaration of Helsinki guidelines. The CONSORT statement checklist was followed for reporting the results presented here.

### 2.2. Experimental Design and Subjects of the Study

This is a pilot cross-over clinical trial using children of both genders, with ages ranging between 4 to 8 years. Children that need at least two dental treatments under the inferior alveolar nerve block were screened. The sample was selected for convenience at Polyclinic Getúlio Vargas, Uberaba University, Uberaba, Minas Gerais, Brazil. Children with spontaneous painful symptoms and guardians who reported some type of chronic disease and/or syndromes, or used medication were excluded.

All children received two types of treatment on different days, and they were randomly allocated the first type of treatment. The two treatments were as follows: local anesthesia performed with a visual passive distraction crocodile case device for carpule (Angie by Angelus^®^, Londrina, PR, Brazil), as demonstrated in Figure 1 (experimental); and local anesthesia performed without the device in the conventional way (control).

The communication, which was a standard set of verbal instructions based on a pre-script given regarding local anesthetic injections including age-appropriate euphemisms with verbal positive reinforcement, was used for both groups. Behavior evaluation was performed according to the Frankl scale (1962) [17,22,23,24]. The anxiety levels were evaluated by measuring heart rate and salivary cortisol levels [23,25].

### 2.3. Administration of the Inferior Alveolar Nerve Block

The local anesthesia procedure was performed using the topical anesthesia technique for both treatment groups. After surface drying, a sterile cotton tip was used to apply a topical anesthetic gel containing 20% benzocaine (Benzotop^®^, DFL, Taquara, RJ, Brazil) to the puncture region. The gel was massaged into the mucosa with moderate pressure for 30 s. After 2 min, the excess topical anesthetic was removed. A nerve block with local anesthetic was performed with a 27-gauge needle, 25 mm long. A total of 1.5 mL of lidocaine-based anesthetic with 2% epinephrine 1:100,000 was injected [6].

A small amount of solution was injected, and, after a negative aspirate, the needle was advanced until bone contact. The administration rate was 1.5 mL of anesthetic in a minimum of 60 s. The needle was slowly withdrawn and when approximately half the length of the needle remained, it was aspirated again. If negative, a portion of the remaining solution was used to anesthetize the lingual nerve. The needle was then withdrawn and held [6]. The standardization of the technique was achieved by the principal investigator after previous training and calibration. The anesthetic procedure was performed in the presence of the parents/guardians.

### 2.4. Evaluate Fear and Anxiety through Behavioral Management

The child’s behavior was observed and categorized according to the Frankl scale (1962) [22]. The description is presented in Table 1. Categorization was achieved by the principal investigator after previous training and calibration.

### 2.5. Evaluate Fear and Anxiety through Physiological Changes

Fear and anxiety were evaluated through physiological changes according to changes in heart rate [24] and salivary cortisol level [23,25]. Heart rate measurement was performed 10 min before and 10 min after the anesthetic procedure. Heart rate measurement was performed using a digital oximeter positioned on the left index finger.

Salivary cortisol levels were also measured 10 min before and 10 min after the anesthetic procedure. The collection was performed with a cotton roll positioned sublingually; approximately 1 mL of saliva was achieved. The free cortisol levels in saliva were determined in duplicates using a high-sensitivity salivary cortisol enzyme immunoassay kit according to the manufacturer’s instructions (Salimetrics, LLC, State College, PA, USA). The samples from each subject were assayed in the same batch.

### 2.6. Statistical Analysis

For statistical analysis, SPSS Statistics (IBM, Armonk, NY, USA) was used. Data normality was verified by Shapiro–Wilk. Wilcoxon test was used to compare the studied variables in the same individual before and after anesthesia. The significance level was set at an alpha of 5%.

## 3. Results

A total of 13 children were included in this pilot study, 8 (61.5%) were girls and 5 (38.5%) were boys. No child was categorized as having definitely negative behavior. Five (38.4%) were categorized with negative behavior, four (30.8%) were categorized with positive behavior, and four (30.8%) were categorized with definitely positive behavior. None of the children dropped out of the study. The characteristics of the sample are shown in Table 2.

Figure 2 shows the influence of the visual passive distraction of the crocodile case device for carpule on heart rate and salivary cortisol levels. The mean heart rate during the treatment with the visual passive distraction device fell from 81.8 to 78.1, while in the control treatment, mean heart rate rose from 91.4 to 100. The mean heart rate during the treatment with the passive distraction device was statistically lower than during the control treatment (*p* < 0.05). The mean salivary cortisol levels during the treatment with the visual passive distraction device fell from 2.0 to 1.6, while in the control treatment, the salivary cortisol levels rose from 2.1 to 2.4. The mean salivary cortisol levels during the treatment with the passive distraction device were also statistically lower than during the control treatment (*p* < 0.05).

## 4. Discussion

Children’s fear and anxiety directly affect routine dental practice [2,3,4]. In addition to the hindered behavior, there is a greater sensitivity and perception of pain in patients with feelings of fear and anxiety [2,5,6,7,8,9]. The current scientific evidence shows satisfactory results from numerous behavioral management protocols for limiting and controlling dental fear and anxiety [2,6,12,13,14]. However, humanized dentistry supports the possibility of applying behavioral management techniques using a non-pharmacological approach, such as passive distraction [3,4,6,17,18]. The present study aimed to evaluate the influence of a new ludic visual passive distraction device, known as the crocodile case for carpule, on the management of fear and anxiety during inferior alveolar nerve block in children. Our results show a statistically significant reduction in heart rate and salivary cortisol levels with the crocodile case for carpule during the inferior alveolar nerve block procedure in children. The null hypothesis was rejected.

Dental fear is defined as a negative reaction to specific threatening stimuli associated with dental treatment [2]. Dental anxiety can be defined as an excessive or harmful negative emotional state experienced by dental patients [2]. It is worth mentioning that both conditions have a multifactorial and complex etiology; several local, environmental, and psychological factors inherent to children and adolescents can be cited as influential [1]. Such evidence further highlights the importance of humanized care. Non-pharmacological behavioral management techniques are simple, safe, and effective when well indicated [15,16].

Visual and/or auditory passive distraction has shown promising results in managing child behavior [3,4,17,18]. The transmission of negative attitudes from parents regarding their fears and anxieties is another issue in the dental office. Several pieces of evidence have demonstrated an association between parental and child fear and anxiety [1,20]; however, this is a limitation of our study because we did not evaluate the caregivers. The local anesthesia procedure is one of the most cited reports of the cause of fear/anxiety in pediatric and adult dental practice [19,20,21]. Parental fear and anxiety may be a common reason to avoid child dental treatment, which over time can result in the further deterioration of oral health, intensifying the need for an anesthetic procedure [26]. This fact also helps with the need to make anesthetic procedures more playful. However, studies that include devices intended for this purpose are still necessary.

The physiology of fear is supposed to be initiated when the child comes into eye contact with what frightens him. Stimuli are sent to the brain, which initiates its reaction to the frightening stimuli. There is an excessive production and endogenous release of adrenaline, noradrenaline, and acetylcholine, which trigger changes in the individual’s body and may even contribute to levels of anxiety [27]. The crocodile case for carpule (Angie by Angelus^®^, Londrina, PR, Brazil) was developed to minimize the fear and anxiety during the anesthetic procedure. The tool is made of flexible rubber, with a playful shape that adjusts and hides the carpule syringe and needle at the time of anesthesia. The flexible rubber also has the advantage of being able to be sterilized.

The release of endogenous catecholamines that causes an increase in heart rate is a widely used indicator in the perception of fear and anxiety [27]. Our results demonstrate a significant decrease in heart rate after the anesthetic procedure performed with the crocodile case for the carpule. Although there are no studies that have evaluated devices such as the crocodile case for carpule specifically, the use of passive visual distraction with other means demonstrates similar results regarding heart rate in children [4,17]. This result especially enhances the aims of this study because of the applicability of a low-cost and effective means of controlling fear and anxiety in pediatric dentistry.

Our study also highlights the heterogeneity of the sample about behavior according to Frankl’s Behavioral Scale (1962) [22]. Such heterogeneity implies that future multicentric studies should be carried out with larger and more homogeneous samples in each sample subset. However, there is a decrease in heart rate in all children who underwent the anesthetic procedure with the crocodile case for carpule, regardless of categorization according to Frankl’s Behavioral Scale (1962) [22]. All children, regardless of categorization according to the Frankl Behavioral Scale (1962) [22], who underwent the anesthetic procedure without the crocodile case for carpule, their heart rate tended to increase, suggesting conditions of fear and anxiety, and praising the effectiveness of the crocodile case for carpule.

Our results demonstrate a decrease in serum levels of salivary cortisol in the group of children undergoing an anesthetic procedure using the tool, suggesting a significant reduction in feelings such as fear and anxiety. Cortisol is a biomarker that has long been used in dental and medical research [22,25,28,29]. Increases in salivary cortisol levels are evidenced in patients with feelings of fear/anxiety since the first dental visit [23,25]. In our descriptive results, in the group of children with negative behavior who underwent the anesthetic procedure without the crocodile case for carpule, the salivary cortisol levels after the anesthesia procedure decreased. It is suggested that the feeling of anxiety was more prevalent in these children, since anxiety is anticipatory, and the manifestation of anxiety is due to uncertainty about a future threat and the inability to mitigate or avoid it.

In our study, the intervention was performed with standardized procedures by a single trained and calibrated operator. Other operators are also necessary for evaluation and adaptation regarding the technique of using the crocodile case for carpule, because of the trust that the child places in the professional. Developmental and cognitive factors and situational conditions can influence a child’s perception of fear and anxiety and influence the results of self-reported scales [30]. In conclusion, although future studies are necessary to complement the scientific evidence regarding the crocodile case for carpule, the authors encourage the use of this visual passive distraction tool in dental procedures that require anesthesia in pediatric dental practice. Future studies with adequate sample numbers to support this pilot study are necessary to confirm the hypothesis in children with definitely negative behavior. However, it is already possible to suggest its applicability in the behavioral management of children with fear and anxiety in dental practice.

## 5. Conclusions

In this pilot study, the crocodile case for carpule as a visual passive distraction device contributed positively to the management of fear and anxiety during inferior alveolar nerve block in children. Future studies using this pilot study as a reference should be performed in different populations and children with different ages and behavior.

## Figures and Tables

**Figure 1 ijerph-20-01793-f001:**
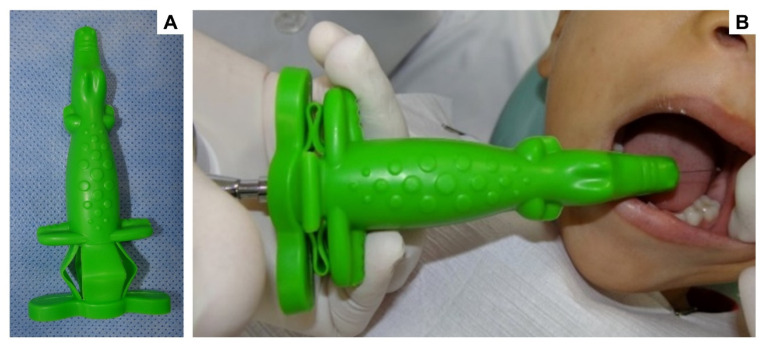
Local anesthesia was performed with a visual passive distraction crocodile case device for carpule (Angie by Angelus^®^, Londrina, PR, Brazil). In (**A**), note the crocodile case for carpule (Angie by Angelus^®^, Londrina, PR, Brazil), and in (**B**), the local anesthesia performed.

**Figure 2 ijerph-20-01793-f002:**
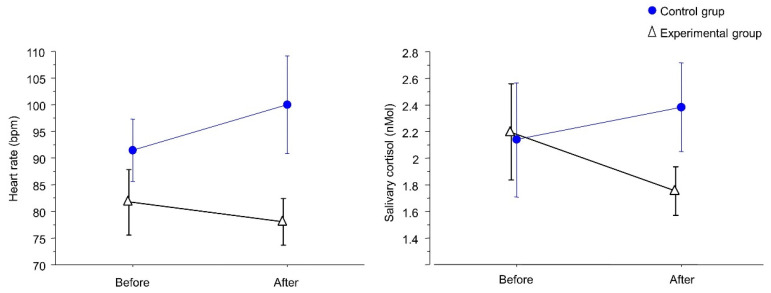
Influence of the crocodile case for carpule, on heart rate and salivary cortisol levels. The points show the mean (±standard deviation) of heart rate and salivary cortisol before and after anesthesia (*p* < 0.05).

**Table 1 ijerph-20-01793-t001:** Frankl behavior rating scale (Frankl et al., 1962) [22].

Rating	Description
Definitely negative	Refusal of treatment, crying forcefully, fearful, or any other overt evidence of extreme negativism.
Negative	Reluctant to accept treatment, uncooperative, or some evidence of negative attitude but not pronounced, i.e., sullen and withdrawn.
Positive	Acceptance of treatment, at times cautious, willingness to comply with the dentist, at times with reservation, but follows the dentist’s directions cooperatively.
Definitely positive	Good rapport with the dentist, interested in the dental procedure, laughing, and enjoying the situation.

**Table 2 ijerph-20-01793-t002:** Characteristics of the sample.

Characteristics		Negative	Positive	Definitely Positive	Total
Sex n (%)					
Girls		2 (15.4)	3 (23.1)	3 (23.1)	8 (61.5)
Boys		3 (23.1)	1 (7.7)	1 (7.7)	5 (38.5)
Treatment with visual passive distractive crocodile case device for carpule					
Before the procedure	Heart rate Mean (SD)	95 (19.8)	78 (18.9)	58.5 (42.3)	81.8 (18.5)
	Min.–Max.	81–109	51–93	55–92	51–109
	Salivary cortisol Mean (SD)	2.7 (1.7)	2.1 (1.3)	1.5 (0)	2.0 (1.1)
	Min.–Max.	1.5–3.9	1.5–4	1.5–1.5	1.5–4
After the procedure	Heart rate Mean (SD)	75 (15.6)	76 (11)	62.3 (44)	78.1 (13)
	Min.–Max.	64–86	62–87	63–97	62–97
	Salivary cortisol Mean (SD)	2.2 (1.2)	1.5 (0.2)	1.4 (0.1)	1.6 (0.5)
	Min.–Max.	1.3–3	1.2–1.7	1.3–1.5	1.2–3
Treatment with normal carpule					
Before the procedure	Heart rate Mean (SD)	100.3 (27.1)	92.5 (14.8)	81.5 (16.9)	91.4 (20.1)
	Min.–Max.	62–120	81–112	61–102	61–120
	Salivary cortisol Mean (SD)	2.9 (2.4)	1.9 (0.7)	1.5 (0)	2.1 (1.6)
	Min.–Max.	1.5–7	1.5–2.9	1.5–1.5	1.5–7
After the procedure	Heart rate Mean (SD)	116.8 (47.9)	97 (25.3)	86.3 (10.7)	100 (31.7)
	Min.–Max.	60–177	61–116	73–99	60–177
	Salivary cortisol Mean (SD)	2.5 (1.3)	2.1 (0.9)	2.6 (1.6)	2.4 (1.2)
	Min.–Max.	1.5–4.9	1.5–3.4	1.5–4.9	1.5–4.9

## Data Availability

Data is contained within the article.

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
