# Peer review of "Effect of the Case for Carpule as a Visual Passive Distraction Tool on Dental Fear and Anxiety: A Pilot Study"

_ijerph, 2023, doi:10.3390/ijerph20031793_

Round 1

Reviewer 1 Report

This manuscript describes a pilot cross-over intervention study where investigators compared a novel device shaped like a crocodile that fits over an anesthetic syringe to hide the syringe and distract children during an inferior alveolar injection. This is a novel idea (although this reader wondered why they chose a crocodile, which might be a bit scary – as opposed to say a bunny which might be less scary), and this paper demonstrates proof-of-concept and provides some data to show that this device could be used to reduce anxiety/fear in children surrounding injections.

The manuscript is well-written and the methods are appropriate for a pilot study. The fact that the n is small actually lends credence to the idea that there was indeed a difference between groups regarding heart rate and cortisol levels. The authors are appropriately cautious in overstating the significance of their findings given that this is a pilot study.

Some suggestions to improve the manuscript:

-        Please consider providing actual values in the test of the results as opposed to only the results of the significant testing (lines 146- 49). It would be helpful, for example, to have a sentence describing the direction of the change, for example:

The mean heart rate during the treatment with the passive distraction device fell from 81.8 to 78.1, while without the treatment it actually rose to 100 from 91.4. This difference between the intervention and control was statistically significant (p<0.05).  

This could be done also for the differneces found in cortisol levels as it was a little hard to go back and forth between the text and Table 2 to find the actual values. The authors may want to consider also adding these numbers to the abstract.

-        It may be helpful to readers to more completely describe the intervention sooner in the manuscript as this reader had no idea what the intervention actually was until page 3. A short description of this intervention as a crocodile-shaped syringe cover would probably suffice.

-        The authors should consider adding a paragraph in the Discussion section that outlines what steps they plan to take to further test this product. Is there a plan for a larger study, one that would include more children who are fearful?

Reviewer 2 Report

This cross-over study aimed to evaluate the effect of a visual passive distraction 18 tool, a case for carpule, in the management of fear and anxiety during invasive dental treatment. In general, the manuscript is well-written and it has the necessary elements to consider a high-quality study.

I have some suggestions and comments in order to improve this manuscript:

1.       Please be sure that the keywords are MeSH terms.

2.       Please mention the strengths and limitations of the study in greater detail

3.       Please mention recommendations for research and dentistry practice in greater detail and derived from the study findings.

4.       It is important to mention if this trial was registered, if not, justify or explain why not. 

5. Please review grammar and writing in the whole text. 

Reviewer 3 Report

The present study deals with a topic that is not particularly original or innovative. There are many studies in this regard in the literature.

-A review of English style and grammar by an expert native speaker is required.

-Why are there no null hypotheses in the "Introduction" section? At least one null hypothesis must be inserted at the end of the "Introduction" section and then resumed at the beginning of the "Discussion" section to assert whether the results of the study have allowed it to be accepted or rejected.

-The authors seem to have paid little attention to the authors' guidelines, in fact, throughout the text there are many cases in which the numbering of the references is not respected as required by the journal. E.g., "Line 158: [2, 5, 6, 7,8,9]." Should be [2, 5-9].

-Have tests been done to evaluate the normal distribution of the data obtained (e.g., Shapiro-Wilk or Kolgomorov-Smirnov tests)?

-It is necessary to insert a more detailed and in-depth paragraph regarding the limits of the research protocol used.

-The "Conclusions" section is unacceptable at present as it is too poor, it needs to be integrated.

-There are too many old references in the bibliographic list. Update it by inserting more recent papers.
